# Diagnostic Accuracy of Ex Vivo Confocal Microscopy for Surgical Margin Assessment of High-Risk Nodular Basal Cell Carcinoma

**DOI:** 10.3390/cancers17183019

**Published:** 2025-09-16

**Authors:** William Stramke, Luca Tonellotto, Emmanuella Guenova, François Kuonen

**Affiliations:** Department of Dermatology and Venereology, Lausanne University Hospital, University of Lausanne, 1011 Lausanne, Switzerland

**Keywords:** basal cell carcinoma, nodular, confocal laser scanning microscopy, Mohs surgery, peripheral and deep en-face margin assessment, surgical margins, intraoperative imaging, skin cancer, histology

## Abstract

With the rising incidence of basal cell carcinoma (BCC), more efficient therapeutic strategies are needed. Although the nodular subtype is typically considered indolent, tumors located in anatomically sensitive areas or exceeding certain size thresholds may present a high risk of recurrence and require adapted surgical approaches that are time- and resource-intensive. As the nodular histological pattern is readily identifiable using of ex vivo confocal laser scanning microscopy (EVCM), our retrospective study aimed to evaluate its suitability for surgical margin assessment in high-risk nodular BCCs. Compared to conventional histopathology, EVCM demonstrated very high diagnostic accuracy. The study also highlighted potential technical limitations of the method. Overall, our findings support the integration of EVCM-assisted surgery into clinical practice, particularly for the management of high-risk nodular BCCs.

## 1. Introduction

Basal cell carcinoma (BCC) is the most common malignancy affecting humans and represents approximately 75% of all diagnosed skin cancers worldwide [1]. Its predominance highlights both the scale of the public health burden and the necessity for efficient and precise diagnostic and therapeutic strategies. In recent decades, the incidence of BCC has shown a consistent upward trend across numerous countries. This increase is commonly attributed to a combination of factors, chiefly the progressive aging of populations and persistent patterns of cumulative sun exposure [2].

Surgical excision remains the cornerstone and gold standard of BCC treatment [1]. The primary aim of surgical intervention is to achieve complete tumor removal while minimizing damage to the surrounding healthy tissue. To confirm complete excision, histopathological assessment of the surgical margins is systematically performed. This microscopic evaluation ensures that no residual cancerous cells are left behind which could otherwise give rise to local recurrence. Indeed, recurrence is a significant concern in dermatologic oncology, particularly for tumors located in high-risk anatomical sites or displaying aggressive growth patterns.

However, conventional histopathological techniques, though widely adopted, are inherently limited. Typically, these methods involve the processing and staining of formalin-fixed tissue sections, using vertical “bread-loaf” sections that sample only a small fraction—commonly between 1 and 2%—of the total surgical margin surface area [3,4,5]. This partial examination leaves a substantial proportion of the margin unchecked, introducing the risk that residual tumor cells may be missed during analysis.

This diagnostic gap has stimulated the implementation of Mohs micrographic surgery for 100% margin control, as well as alternative forms of peripheral and deep en-face margin assessment (PDEMA) approaches. Unlike conventional techniques, PDEMA methods permit comprehensive, 3D examination of the surgical margin, thereby maximizing the likelihood of complete tumor clearance [6]. Indication include tumors larger than 2 cm in diameter, those located in functionally or cosmetically sensitive areas (i.e., the face, ears, hands, feet, pretibial region, and anogenital region; any size), lesions with poorly defined clinical borders, those showing histologically aggressive subtypes (e.g., infiltrative, sclerodermiform, micronodular, basosquamous), or recurrent tumors [1]. Although highly effective, with recurrence rates among the lowest of all skin cancer treatments [7], PDEMA approaches based on conventional histopathology are resource-intensive, both in terms of cost and time. Moreover, they require dedicated surgical expertise, immediate access to histopathological facilities, and are not widely available in all clinical settings.

In light of the rising incidence of BCC (including high-risk BCC), optimizing the efficiency of PDEMA techniques without diminishing their diagnostic accuracy has become increasingly important. In recent years, ex vivo confocal laser scanning microscopy (EVCM) has emerged as a valuable optical imaging technique capable of generating high-resolution images of freshly excised, unfixed tumor specimens [8]. Unlike conventional histopathology, which requires formalin fixation, paraffin embedding, sectioning, and staining, EVCM enables rapid visualization of tissue architecture in real time, significantly reducing turnaround time for intraoperative or immediate post-excision analysis. This method has been successfully applied to a variety of skin tumors, including BCC [9,10], squamous cell carcinoma [11,12] and dermatofibrosarcoma protuberans [13]. One of its key advantages lies in its ability to provide a comprehensive assessment of surgical margins—like conventional PDEMA approaches—without altering or destroying the specimen, thereby preserving tissue integrity for potential subsequent histological or molecular analysis.

The diagnostic accuracy of EVCM has been assessed in numerous studies, particularly in the context of BCC. Although its application shows promise, the reported diagnostic performance remains variable, with sensitivity values ranging from 73% to 100% and specificity ranging from 89% to 100% [10,14,15,16,17,18,19,20]. These variations are likely attributable to several interrelated factors. The quality of image acquisition and the accuracy of interpretation depend heavily on the specific EVCM device employed, the operator’s level of expertise, and the anatomical location of the tumor. Indeed, skin thickness, surface texture, and curvature can all influence tissue flattening and, consequently, image clarity. In addition, the histological subtype of the BCC significantly impacts detectability. Nodular BCCs, characterized by well-defined borders and densely packed tumor nests, are generally more readily identified. In contrast, aggressive subtypes—such as infiltrative and sclerodermiform BCCs—tend to display subtle, ill-defined, and diffusely infiltrative growth patterns, which pose greater diagnostic challenges when using EVCM [16,20].

In this study, we questioned the diagnostic performance of EVCM compared to H&E-stained paraffin embedded sections for PDEMA of nodular BCCs at high risk of recurrence.

## 2. Materials and Methods

### 2.1. Study Design and Setting

This retrospective, single-center observational study was conducted at Lausanne University Hospital (CHUV) between March 2024 and May 2025 to evaluate the diagnostic performance of EVCM (index test) for detecting incomplete excision of high-risk nodular BCC. The reference standard was conventional paraffin-embedded haematoxylin and eosin (H&E) histopathology, processed within the PDEMA workflow. The study protocol complied with the Declaration of Helsinki (1975, revised in 1983) and was approved by the Institutional Review Board of Lausanne University Hospital and the local ethics committee (protocol number 2023-02206). Written informed consent was obtained from all participants. Eligible participants were adults (>18 years) with a clinical and histopathological diagnosis of nodular BCC who underwent EVCM-assisted surgery at CHUV. Patients were excluded if the diagnosis was uncertain, or if clinical or histopathological documentation was incomplete. From the assembled margin specimens associated with the identified patient cohort, we excluded those for which matched EVCM and H&E-based PDEMA reports were not available (Figure 1). Collected data included patient age, sex, tumor location, tumor size, EVCM reports, and corresponding PDEMA histopathological results.

Confocal digital images were obtained and assessed intraoperatively by the operating dermatosurgeon (LT or FK), prior to availability of histopathology-based PDEMA results. BCC-specific criteria included well-demarcated fluorescent areas with increased nuclear density, peripheral palisading, clefting, and nuclear polymorphism [8]. To facilitate digital image interpretation, the operating dermatosurgeons (LT or FK) used the zoom, contrast adjustment, and purple (H&E-like) digital rendering modes available on the VivaScope^®^ 2500 system, Vivascan Version 11.0.r1206 (VivaScope GmbH, Münich, Germany). Live Z-stacking helped discriminate supsicious BCC foci from adnexal structures like hair follicles or sebaceous glands. On EVCM, margins were classified as positive when suspicious BCC foci were identified at the external surface of peripheral margin specimens or at the bottom surface of deep margin specimens. Subsequent histopathological evaluation of paraffin-embedded PDEMA sections was performed by a dermatopathologist not involved in the surgical procedure (EG). In PDEMA, margins were classified as negative when the most external representative sections of peripheral specimens or the deepest representative sections of deep specimens were found to be free of tumor. Although blinding to the EVCM results was intended, partial unblinding could not be fully excluded due to potential access to the EVCM report by the dermatopathologist or specimen labeling that might have implied EVCM findings. Discordant cases between EVCM and H&E-based PDEMA results were jointly reviewed and analyzed by FK and WS.

### 2.2. Material

Fifty-one patients fulfilled the study criteria. Their ages ranged from 51 to 96 years (median 75 years), with 47% male and 53% female patients. Tumor size ranged from 5 to 45 mm (median 9 mm). Tumor locations included nose or peri-nasal (*n* = 19), temple or lateral forehead (*n* = 15), ear or peri-auricular (*n* = 7), perioral (*n* = 5), mandible or lower cheek (*n* = 2), scalp (*n* = 2), and eyelid or peri-orbital (*n* = 1). All tumors were considered high-risk according to inetrnational guidelines [1] based on anatomical location or, in the case of the two scalp lesions, on tumor size (Appendix A). One lesion represented a recurrent BCC. From the total number of margin specimens obtained from the 51 patients, 171 fulfilled the study criteria, comprising 122 peripheral margin specimens and 49 deep margin specimens.

### 2.3. Quantification and Statistical Analysis

The primary outcome measure was the diagnostic performance of EVCM compared with the reference standard, expressed as sensitivity, specificity, positive predictive value, and negative predictive value. Estimates were calculated overall and stratified by specimen type. Ninety-five percent confidence intervals were computed using GraphPad Prism software, Version 10.2.0.

### 2.4. EVCM- and Conventional Histopathology-Based PDEMA Workflow

EVCM was performed using VivaScope^®^ 2500, a confocal laser scanning microscope equipped with two laser sources: a 785 nm diode laser for reflectance imaging and a 488 nm laser for fluorescence imaging, providing a lateral resolution of approximately 1 µm, an axial resolution of ~5 µm, and an imaging penetration depth of up to 200 µm [21,22]. For EVCM-assisted PDEMA, peripheral margins were excised using a 2 mm wide double blade, cut into segments of up to 2 cm if needed to best fit the VivaScope^®^ 2500 imaging window, and imaged for their external side. Central deep margin specimens were obtained with a single blade, cut into segments of up to 2 cm^2^ if needed to best fit the VivaScope^®^ 2500 imaging window, and imaged for their bottom side. Freshly excised, unfixed tissue specimens were gently blotted to remove surface moisture and excess blood. Specimens were first immersed for 10 s in an aqueous solution of acridine orange at a concentration of 1.0 mM (Morphisto GmbH, Offenbach, Germany), followed by immersion in a 0.5% fast green FCF solution (Morphisto GmbH, Offenbach, Germany) for 5 s. A brief rinse in 70% ethanol was performed prior to a second, similar staining cycle. Following staining, specimens were gently rinsed in PBS, blotted, and mounted within a VivaScope^®^ cassette with magnetized glasses and a foam support, ensuring optimal tissue flattening for high-quality confocal imaging. EVCM scanning required approximately 2 min for 2 cm^2^ field-of-view. After EVCM imaging, specimens were fixed in 4% formaldehyde and processed for conventional PDEMA via standard en-face histological techniques following a technique similar to the reported “Tuebinger Torte” method [23].

## 3. Results

A total of 171 surgical margins were analyzed, including 122 peripheral and 49 deep margins. Using conventional, histopathology-based PDEMA as the reference standard, EVCM correctly identified 15 true positive margins (11 peripheral and 4 deep; illustrated in Figure 2, patients #2 and #3) and 153 true negative margins (108 peripheral and 45 deep; illustrated in Figure 2, patient #1). Two false positive cases and one false negative case were observed, all located within peripheral margins. Based on these results, the overall sensitivity of EVCM for the detection of high-risk, nodular BCC was 93.8% (95% CI: 71.7–98.9%), with sensitivities of 91.7% (95% CI: 61.5–98.7%) and 100.0% (95% CI: 51.0–100.0%) for peripheral and deep margins, respectively. The overall specificity reached 98.7% (95% CI: 95.2–99.7%), including 98.2% (95% CI: 93.5–99.6%) for peripheral margins and 100.0% (95% CI: 92.1–100.0%) for deep margins. The overall positive predictive value (PPV) was 88.2% (95% CI: 63.6–97.4%), with values of 84.6% (95% CI: 54.6–96.9%) for peripheral margins and 100.0% (95% CI: 51.0–100.0%) for deep margins. The negative predictive value (NPV) was 99.4% overall (95% CI: 96.4–99.9%), including 99.1% (95% CI: 94.9–99.9%) for peripheral margins and 100.0% (95% CI: 92.1–100.0%) for deep margins. These data are summarized in Table 1.

We analyzed the discrepancies observed between EVCM and conventional histopathology-based PDEMA. The false-negative case attributed to EVCM involved the peripheral margin of a nodular BCC of the eyelid. Comparison with adjacent H&E-stained sections revealed residual BCC located in the superficial dermis, which, on confocal imaging, was partially concealed by a folding of the stratum corneum, complicating detection (Figure 3, patient #1). One false-positive case attributed to EVCM was observed at the peripheral margin of a nodular BCC of the lateral forehead. Comparison with adjacent H&E sections revealed localized benign infundibular extensions displaying some degree of peripheral palisading, potentially mimicking BCC on confocal imaging (Figure 3, patient #2). The second false-positive case was detected in the dermis of a nodular BCC of the ear. Review of the corresponding H&E-stained sections revealed poor histological quality and possibly missing regions, limiting the ability to establish a definitive correlation (Figure 3, patient #3).

## 4. Discussion

Our findings demonstrate that EVCM provides very high diagnostic performance for the detection of nodular BCC when integrated into a peripheral and deep en-face margin assessment (PDEMA) workflow. It supports the clinical value of EVCM as a reliable intraoperative diagnostic tool for margin control in PDEMA surgery of high-risk nodular BCC.

Although not classified as high-risk per se—unlike infiltrative, micronodular, or other histologically aggressive subtypes—nodular BCCs remain the most prevalent form of BCC [24,25] and can nonetheless present a high risk of recurrence, particularly when located in anatomically sensitive or functionally critical areas. Tumor size, clinical margins, and proximity to vital structures may further elevate the recurrence risk, thereby justifying the use of more advanced surgical strategies [1]. In this context, high-risk nodular BCCs constitute a frequent and appropriate indication for Mohs or alternative PDEMA approaches, which allow for precise, margin-controlled excision while preserving surrounding healthy tissue. Our study highlights the potential utility of EVCM as a complementary tool for margin assessment in the surgical management of these lesions.

The reported sensitivity of 93.8% is slightly higher than that observed in two previously published large-scale studies comparing EVCM to Mohs or PDEMA histopathology in BCC assessment [16,18]. This difference likely stems from our selective inclusion of nodular BCCs only, explicitly excluding infiltrative subtypes based on preoperative biopsy results. Nodular BCCs are typically easier to detect due to their well-circumscribed histological architecture and dense tumor aggregates [26]. In contrast, infiltrative BCCs are composed of thin, irregular strands of tumor cells that may elude detection under EVCM, thus reducing sensitivity in heterogeneous study cohorts [16,20]. One prior study reported 100% sensitivity for detecting BCC on the eyelid using EVCM [17]. However, this finding must be interpreted in light of the comparator used: vertical histological sectioning, a technique known to have limited sensitivity due to its restricted sampling of the total margin surface [3,4,5].

Consistently, our study demonstrated a very high negative predictive value (NPV, 99.4%), indicating a low likelihood of residual tumor being missed during intraoperative assessment. Specifically, the NPV for deep margins reached 100%, suggesting excellent reliability in ruling out tumor extension at this level. While this result should be interpreted with caution—given the relatively low incidence of positive deep margins in our cohort and wide confidence interval—it reflects the relative ease of detecting nodular BCC in the deep dermis and subcutaneous tissue, where adnexal structures are sparse. This observation, previously described in the literature [19], holds clinical significance, as deep margin recurrences are more difficult to detect and often necessitate more invasive secondary interventions [27].

Direct comparison between EVCM images and adjacent H&E-stained sections provides valuable technical insights. The false negative finding on EVCM was attributed to a folding of the stratum corneum over the superficial dermis (Figure 3, patient #1)—an artifact that has been consistently reported in previous EVCM studies [28], along others like suboptimal flattening, halos or bleaching [14,18]. Such artifacts can compromise interpretation but are typically preventable through careful sample handling. Specifically, ensuring adequate flattening of the tissue prior to imaging and, when necessary, applying gentle pressure to unfold superficial layers can significantly improve image quality and diagnostic accuracy [29]. However, false-negative cases attributed to EVCM may, in reality, reflect false-positive findings from conventional histopathology. It is plausible that tumor strands, although located in close proximity to the surgical margin, do not actually infiltrate it. During conventional histopathological processing, whether in Mohs or PDEMA protocols, a small portion of tissue is typically lost when preparing optimal sections. This technical loss may lead the histotechnician to section directly through the tumor mass rather than at the true surgical edge, thereby suggesting margin involvement even when the margin was histologically clear. By contrast, EVCM requires no tissue sectioning. The fresh specimen is applied directly to the microscope stage and scanned as a whole, preserving the integrity of the entire surface. As a result, the images obtained accurately represent the true peripheral and deep surgical margins without introducing artifacts related to sectioning loss. In clinical practice, this could reduce overtreatment due to overestimation of margin involvement, particularly in high-risk BCCs where tissue preservation is essential.

Our findings indicate suboptimal positive predictive value (PPV, 88.2%). This suggests that the use of EVCM may lead to unnecessary tissue resection, particularly at peripheral margins where histological structure mimicking BCC are mostly encountered (Figure 3, patient #2). However, this interpretation must be weighed against the intrinsic limitations of the histological gold standard, which may fail to identify focal tumor deposits detectable by EVCM [30] and thus result in an overestimation of false positives attributed to this technique. Indeed, conventional H&E-based histological processing (for Mohs or PDEMA approaches) is subject to a number of technical constraints that can impair the accurate evaluation of surgical margins. These include sectioning artifacts, tissue loss, suboptimal sampling, and incomplete representation of the resection plane detection (Figure 3, patient #3) [31]. By contrast, EVCM allows for immediate, en face visualization of the entire surgical margin, thereby reducing the likelihood of such artifacts and enhancing the reliability of intraoperative margin assessment.

### Study Limitations

This study has several limitations. A major limitation is that our analysis focused exclusively on histopathological comparison rather than on long-term clinical outcomes such as local recurrence, which would require prospective clinical follow-up studies. Additionally, the comparison between EVCM and conventional, micrographic H&E-stained slides is inherently constrained by the use of adjacent—rather than identical—tissue sections. Consequently, small tumor foci present in one plane may be absent in the other, potentially leading to apparent discrepancies in margin status. This spatial mismatch may contribute to an overestimation of both false negative and false positive findings attributed to EVCM. Also, its single-center design may limit external validity, the influence of reader expertise and the associated learning curve were not examined, and inter-observer variability was not assessed. This study focused on nodular BCC and did not consider more aggressive BCC subtypes, which might be more challenging to diagnose on EVCM. Future work should also explore the potential implementation of deep learning algorithms in the clinical workflow, as it may further enhance diagnostic accuracy and reproducibility [32].

## 5. Conclusions

EVCM demonstrates very high diagnostic performance within the PDEMA workflow for high-risk nodular BCC and thus represents a valuable tool to support intraoperative margin assessment. Our findings support the broader implementation of EVCM-assisted surgery in the near future, although additional studies are required to refine its precise indications relatively to Mohs surgery or conventional PDEMA, particularly with regard to BCC histological subtype and anatomical location. Finally, its clinical utility should be further validated in prospective studies with long-term clinical follow-up.

## Figures and Tables

**Figure 1 cancers-17-03019-f001:**
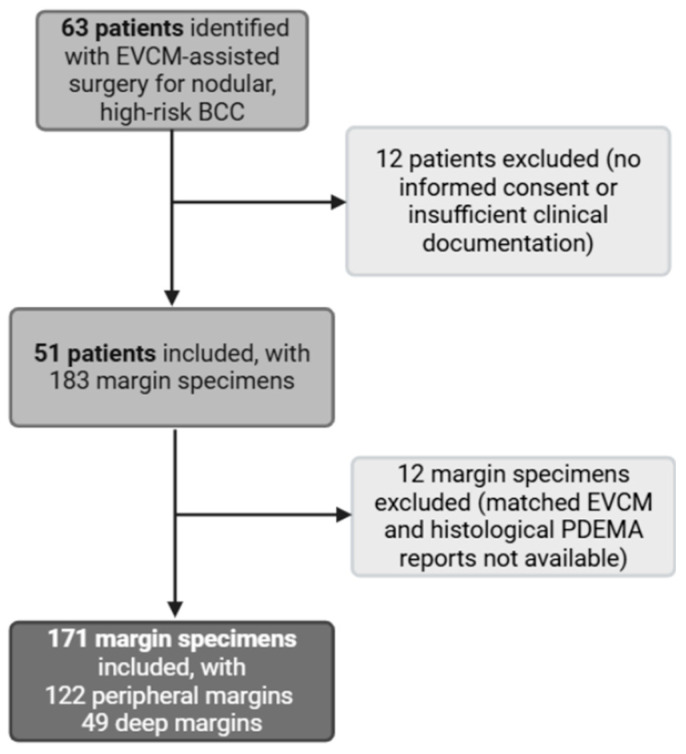
Study flowchart. EVCM: ex vivo confocal laser scanning microscopy. PDEMA: peripheral and deep en-face margin assessment.

**Figure 2 cancers-17-03019-f002:**
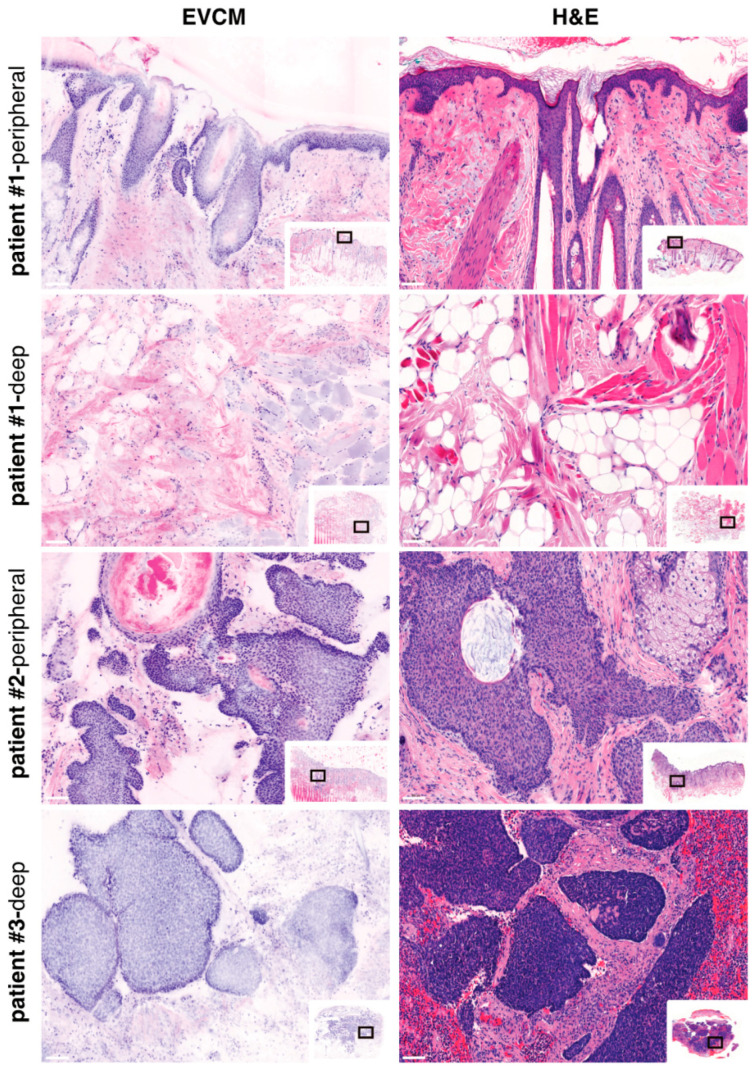
Ex vivo confocal laser scanning microscopy (EVCM) image (**left**) and corresponding H&E staining (**right**) of PDEMA surgical specimens. Patient #1 illustrates the EVCM “true-negative” case, for both peripheral and deep margins; Patients #2 and #3 illustrate the EVCM “true-positive” cases, for peripheral and deep margins, respectively. Scale bars indicate 100 µm.

**Figure 3 cancers-17-03019-f003:**
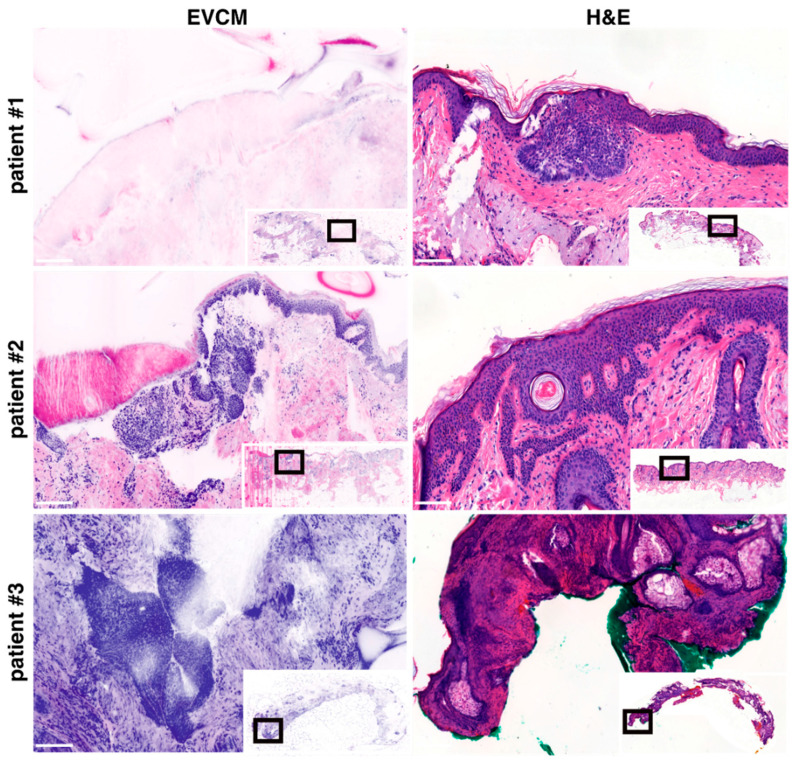
Ex vivo confocal laser scanning microscopy (EVCM) images (**left**) and corresponding H&E stainings (**right**) of basal cell carcinoma (BCC) patients. Patient #1 illustrates the EVCM “false-negative” case; Patients #2 and #3 illustrate the EVCM “false-positive” cases. Scale bars indicate 100 µm.

**Table 1 cancers-17-03019-t001:** Diagnostic performance of EVCM using VivaScope^®^ 2500 compared to histologic analysis on paraffin-embedded H&E sections for PDEMA of high-risk, nodular BCCs. CI; confidence interval.

	All	Peripheral Margins	Deep Margins
True positive (*n*)	15	11	4
False positive (*n*)	2	2	0
True negative (*n*)	153	108	45
False negative (*n*)	1	1	0
Sensitivity (%)	93.8 (95% CI: 71.7–98.9)	91.7 (95% CI: 61.5–98.7)	100.0 (95% CI: 51.0–100.0)
Specificity (%)	98.7 (95% CI: 95.2–99.7)	98.2 (95% CI: 93.5–99.6)	100.0 (95% CI: 92.1–100.0)
Positive predictive value (%)	88.2 (95% CI: 63.6–97.4)	84.6 (95% CI: 54.6–96.9)	100.0 (95% CI: 51.0–100.0)
Negative predictive value (%)	99.4 (95% CI: 96.4–99.9)	99.1 (95% CI: 94.9–99.9)	100.0 (95% CI: 92.1–100.0)

## Data Availability

The data that support the findings of this study are available from the corresponding author upon reasonable request.

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
