# Peer review of "Diagnostic Accuracy of Ex Vivo Confocal Microscopy for Surgical Margin Assessment of High-Risk Nodular Basal Cell Carcinoma"

_cancers, 2025, doi:10.3390/cancers17183019_

Round 1

Reviewer 1 Report

Comments and Suggestions for Authors

General commets

  • Very interesting work and results.
  • In overall, the methods should be clarified, and language edited.

Summary

  • Edit accordingly to the comments given further.

Abstract

  • Edit accordingly to the comments given further.

Background

  • The whole background can be shortened, repeating should be avoided and text flow should be more straightforward.
  • In background it should be more clearly presented that there is Mohs micrographic surgery (MMS) for 100% margin control as classical method for high-risk BCCs, and as option there exists other 3D histopathological methods to assess 100% of margins. Now reader gets easily confused which method is used in this study.
  • Please, be concise in abbreviations, unnecessary repetition of the long word and putting abbreviation in parentheses repeatedly.
  • Would it be more convenient to use EVCM as abbreviation for “ex vivo confocal microscopy” instead of saying ex vivo CLSM, as CLSM can refer to in vivo or ex vivo, and EVCM would be more precise term.
  • “In this study, we evaluated the performance of ex vivo CLSM within a clinical workflow for surgical margin assessment of nodular BCCs at high risk of recurrence due to anatomical location or tumor size, and thus indicated for micrographic surgery. Confocal digital images were compared to matched, adjacent paraffin-embedded, hematoxylin and eosin (H&E)-stained sections.” -> only say the aim, the describing of methods belongs to method chapter
  • For example say ” In this study, we evaluated the performance of EVCM compared to H&E-stained paraffin embedded sections in 100% surgical margin assessment of nodular BCCs at high risk of recurrence due to anatomical location or tumor size.”

Materials and Methods

  • Please clarify the whole materials and methods chapter.
  • Now the content of the paragraphs do not correspond the subtitles.
  • For example subtitle 2.1 should be “Study design and settings”, where is clearly said that this was a retrospective observational single centre study, where the diagnostic accuracy of EVCM in surgical margin assessment of high-risk nodular BCCs was compared to conventional H&E histopathology in PDEMA workflow. Then the study place and time is represented with information following Declaration of Helsinki and approval of ethical committee like written, and then inclusion and exclusion criteria are presented.
  • The subtitle 2.2 should be “Materials”
  • The statistical analysis should be a separate subtitle with paragraph describing, which statistical methods were used. Now it is written in subtitle 2.2, but the actual paragraph does not tell anything about the used statistical methods.
  • The paragraph 2.3 could be named “Protocol” or “Workflow”, the procedure is incorrect/unsuitable term.
  • Please clarify, were the tubular strips divided in pieces, which side of the strip was imaged?
  • Please clarify, was the bottom of central piece imaged with CLSM and then sectioned horizontally for H&E?
  • Please be more precise and clarify, the paragraph 2.3 says ”Confocal digital images were evaluated in real time by the operating dermatosurgeons (LT and FK)”, which is prospective action, but earlier is said that the design is retrospective.

Results

  • Clearly presented results.
  • Please specify where the 122 lateral margins comes from, if there were 51 patients, and from 51 tumors was taken a tubular strip with double scalpel.
  • Why only 49 deep margins were assessed, if there was 51 patients?

Discussion

  • Good discussion and comparison to earlier studies.
  • It should be clarified what the writers think: is this workflow design meant to be used only for high-risk nodular BCCs or applied to other aggressive growth-patterns as well? Meant to replace traditional Mohs or be optional?

Limitations

  • One should mention that the material consists only nodular high-risk BCCs and applying this technique to other aggressive growth-pattern BCCs is unclear based on this study.

Conclusions

  • One could mention that same study design should also be studied in aggressive growth pattern high-risk BCCs, comparing CLSM to PDEMA or Mohs.
Comments on the Quality of English Language

The language should be edited in entire manuscript.

Reviewer 2 Report

Comments and Suggestions for Authors

This single‑center retrospective diagnostic‑accuracy study compares ex vivo confocal laser scanning microscopy (CLSM) against paraffin H&E histology for peripheral and deep en‑face margin assessment (PDEMA) in high‑risk nodular BCC. The authors include 51 patients and 171 margins, and report high performance for CLSM overall (sensitivity 93.8%, specificity 98.7%, NPV 99.4%), with particularly strong performance for deep margins (all point estimates 100%; wide CIs). The manuscript is clinically relevant and well‑motivated; CLSM could streamline micrographic workflows where available. However, there are important issues in reporting, internal consistency, statistics, and referencing that must be resolved to meet Cancers’ standards for diagnostic‑accuracy papers (ideally aligning with STARD).

Major comments (requiring author action/clarification)

1) Align reporting with STARD and clearly define index test & reference standard

Please add a brief STARD‑style methods summary (index test, reference standard, flow of patients/specimens, blinding, handling of indeterminate results). Right now, key elements are implied but not fully specified (e.g., whether CLSM calls were recorded prospectively before histology, and how disagreements were handled). A simple participant/specimen flow diagram would help readers follow inclusions/exclusions and the derivation of 171 margin‑level observations.

2) Patient selection and “high‑risk” criteria need a transparent breakdown

You state inclusion of “high‑risk nodular BCCs” treated by micrographic excision. Please tabulate the exact high‑risk criteria met for each case (e.g., location in H‑zone, size threshold, ill‑defined borders, recurrence), and whether tumors were primary vs recurrent. As written, tumor size median 10 mm seems below many size‑based risk thresholds; presumably most cases are “high‑risk” by location—please confirm. Also, the anatomical site counts sum to 48, not 51 (nose 20, forehead/glabella 15, ear 4, lip 4, cheek/preauricular 3, temple 1, eyelid 1). Please correct and reconcile this discrepancy.

3) Inconsistency between text and Table 1 regarding false positives

In Results, you write that all misclassifications (2 FP, 1 FN) were “located within peripheral margins.” Yet the discrepancy narrative describes a second false‑positive “detected in the deep dermis” of an ear BCC (Figure 2, patient #3), while Table 1 lists 0 deep FPs (p. 6). Please resolve this conflict across text, figure legend, and Table 1. If the case was ultimately adjudicated as peripheral rather than deep—or indeterminate—say so plainly and ensure counts match.

4) Define margin positivity and decision rules for CLSM

Please explicitly define what constitutes a positive margin on CLSM and on histology (e.g., any tumor at the inked surface? within a given distance? how were “close” findings handled?). The CLSM criteria are listed (palisading, clefting, etc.), but a thresholding rule for calling a margin positive is not stated. Also clarify who read the CLSM (both dermatosurgeons jointly or independently?), whether readings were consensus or single‑reader, and how discordances (if any) were resolved.

5) Blinding and independence

You state that paraffin sections were read by an independent dermatopathologist blinded to CLSM (EG). Because EG is also a co‑author, please clarify what “independent” means here (independent of the surgical team but part of the study team?) and confirm blinding integrity (e.g., access to CLSM results, grossing notes, or specimen maps). Also, “conventional micrographic H&E histopathological analysis (EG)” is ambiguous—please rewrite so readers understand precisely who did what.

6) Device and protocol details (reproducibility)

The staining section mentions a “second staining cycle” but no durations are given for the second cycle; please specify. Consider adding basic acquisition parameters important for reproducibility: software version, mosaic size/field‑of‑view, typical scan time per specimen, and whether flattening aids were used (e.g., glass weight, adhesive, filter paper), since you rightly attribute a false‑negative to stratum corneum folding (Figure 2, patient #1).

7) Efficiency claims should be supported or softened

The Discussion and Simple Summary suggest CLSM may improve efficiency intraoperatively, but no timing data are reported. Either (a) provide descriptive timing already available from logs (e.g., median scan/read time per margin) or (b) soften the claim to “may improve efficiency” as a hypothesis to be tested, and keep the emphasis on diagnostic accuracy, which you have measured.

8) Interpret the deep‑margin “all 100%” estimates cautiously

Although point estimates for deep margins are 100%, the sample of positive deep margins is small (n=4) and CIs are wide (e.g., sensitivity 51–100%). Please highlight this explicitly in the Discussion to avoid over‑interpretation.

9) Expand and balance the limitations

The limitations section is good but should also acknowledge: single‑center design, reader expertise/learning curve not examined, no inter‑observer variability, margin‑level analysis with clustering, and selection limited to nodular subtype (limits generalizability to infiltrative/morpheaform BCC). These are interpretive caveats, not calls for new experiments.

10) References: fix mismatches and add missing guideline citation

  • The text cites NCCN v2.2025 multiple times but there is no full NCCN reference in the list; please add the formal guideline citation.
  • Ref [23] is used to support the “Tübingen Torte” technique in Methods; the cited 1999 systematic review is not the canonical reference for the Tübingen cake method. Please replace with an appropriate primary method reference.
  • Ref [7] (Mohs for melanoma) is not suitable to support statements about BCC recurrence and could mislead; please cite BCC‑appropriate Mohs literature.

12) Figures and table: correct typos and harmonize terminology

  • Figure 1 (p. 5) labels contain typos (“periperhal”, “deeep”) and the left column header uses “EVCM” while the text uses “CLSM”—please pick one acronym and use it consistently.
  • Figure 2 (p. 6) plus the related text must match the final adjudication of FP/FN cases (see Major Comment 3).
  • Table 1 mixes decimal comma and point (e.g., 91,7 vs 93.8) and spacing around “CI :”; please standardize to journal style (period decimals and “95% CI:”).

13) Further discussion of deep learning techniques in combination with CLSM

Please discuss deep learning techniques (e.g., https://pubmed.ncbi.nlm.nih.gov/36143256/) that can be combined with CLSM

Minor comments (language, formatting, clarity)

  1. Demographics: Instead of “male‑to‑female ratio 0.89,” please provide n (%) by sex and clarify any age distribution (mean ±â€¯SD or median [IQR]).
  2. Specimen handling: State inking/orientation details for PDEMA (colors, mapping) and whether adnexal structures or sebaceous lobules posed frequent CLSM mimics (you mention benign infundibular extensions once—consider a short paragraph on common pitfalls and how readers mitigated them).
  3. Simple Summary: It currently states that integration of CLSM “improves efficiency”; temper to “may improve efficiency” unless you add descriptive timing.

Reviewer 3 Report

Comments and Suggestions for Authors

Research presented in manuscript needs more scientific soundness.

  1. Statistical parameters are not included in the final manuscript. It should be included in separate section
  2.  Graphical representation of data analysis is missing
  3. Micrograph resolution is poor so, difficult to interpret
  4. Discussion of results is inadequate
  5. Conclusion should be elaborated with future perspective

Round 2

Reviewer 1 Report

Comments and Suggestions for Authors

Revision is done appropiately. Now, methods are descriped that reader can understand what has been done in the study.

Reviewer 2 Report

Comments and Suggestions for Authors

The authors have adequately addressed all issues raised

Reviewer 3 Report

Comments and Suggestions for Authors

Authors have replied to all the comments. Revised manuscript can be accepted for publication